# A Comparison between Cigarette Topography from a One-Week Natural Environment Study to FTC/ISO, Health Canada, and Massachusetts Department of Public Health Puff Profile Standards

**DOI:** 10.3390/ijerph17103444

**Published:** 2020-05-15

**Authors:** Risa J. Robinson, S. Emma Sarles, Shehan Jayasekera, Aziz al Olayan, A. Gary Difrancesco, Nathan C. Eddingsaas, Edward C. Hensel

**Affiliations:** Department of Mechanical Engineering, Rochester Institute of Technology, Rochester, NY 14623, USA; ses9066@g.rit.edu (S.E.S.); gbj6142@g.rit.edu (S.J.); aaa4858@rit.edu (A.a.O.); agdpci@cis.rit.edu (A.G.D.); ncesch@rit.edu (N.C.E.); echeme@rit.edu (E.C.H.)

**Keywords:** cigarette, topography, puff flow rate, puff duration, puff volume, interpuff interval, natural environment, FTC/ISO, standards, puff profiles, emissions, MDPH, HC

## Abstract

Standardized topography protocols for testing cigarette emissions include the Federal Trade Commission/International Standard Organization (FTC/ISO), the Massachusetts Department of Health (MDPH), and Health Canada (HC). Data are lacking for how well these protocols represent actual use behavior. This study aims to compare puff protocol standards to actual use topography measured in natural environments across a range of cigarette brands. Current smokers between 18 and 65 years of age were recruited. Each participant was provided with a wPUM™ cigarette topography monitor and instructed to use the monitor with their usual brand cigarette *ad libitum* in their natural environment for one week. Monitors were tested for repeatability, and data were checked for quality and analyzed with the TAP™ topography analysis program. Data from *n* = 26 participants were analyzed. Puff flow rates ranged from 17.2 to 110.6 mL/s, with a mean (STD) of 40.4 (21.7) mL/s; durations from 0.7 to 3.1 s, with a mean (STD) of 1.5 ± 0.5 s; and volumes from 21.4 to 159.2 mL, with a mean (STD) of 54.9 (29.8) mL. Current topography standards were found to be insufficient to represent smoking across the wide range of real behaviors. These data suggest updated standards are needed such that emissions tests will provide meaningful risk assessments.

## 1. Introduction

Tobacco product emissions testing is carried out on mechanical puffing machines using specified topography parameters such as puff volume, puff duration, puff flow rate, and interpuff gap. Standardized topography parameters are needed to compare emission results across different products. Although there are protocols commonly used in scientific labs and industry, there are no well-accepted topography standards for combustible cigarettes, nor have any of the commonly used protocols been shown to represent realistic puffing. Since exposure to harmful and potentially harmful constituents depends heavily not only on the product but also on smokers’ use behavior, a better understanding of how commonly used protocols reflect actual use behavior is needed.

Originally, in 1967, the industry adopted what was called the Cambridge Filter Method, also known as the FTC/ISO protocol (Federal Trade Commission/International Standards Organization) [1,2,3]. The FTC/ISO protocol was deemed obsolete in 2008 because it failed to represent the way smokers actually smoked their cigarettes [4]. Prior to 2008, many products were labeled as low yield because they produced fewer toxicants when mechanically puffed using the FTC/ISO standard. However, when these purportedly low-yield products were used by actual smokers in their natural environment, smokers would take larger, longer, and more frequent puffs [5,6,7]. Smokers were not necessarily being exposed to fewer toxicants when compared to regular cigarette smokers [8].

Today, there are no regulations requiring specific topography standards to be used for machine emissions testing. The Food and Drug Administration’s (FDA) draft guidance issued in 2012 recommends using the FTC/ISO protocol to represent non-intense puffing, and another protocol, known as the Health Canada Test Method (HC), to represent intense puffing [9]. There is also a third commonly used protocol, known as the Massachusetts Department of Public Health protocol (MDPH), which is typically considered a medium-intensity protocol [10]. To date, there have been no data to justify any of these three standards as representing actual use behavior in either the low-, medium- or high-intensity regimes.

The FTC/ISO protocol specifies a 35 mL, 2-s puff with a 60-s interval between puffs. The MDPH protocol specifies a 2-s puff but increases the puff volume from 35 to 45 mL, decreases the interval between puffs from 60 to 30 s and adds the requirement that 50% of the ventilation holes must be blocked [10]. Thus, MDPH is considered to be a more intense puffing regime compared to the FTC/ISO protocol. The HC protocol specifies a 2-s puff, but further increases intensity over the MDPH protocol by increasing puff volume from 45 to 55 mL, and adding the requirement that 100% of ventilation holes must be blocked [11].

Plenty of the current data available for cigarette emissions across numerous manufacturers and brands were collected using the FTC/ISO protocol [12,13]. Some emissions data [14] and in vitro toxicity data [15,16] were collected using the MDPH and HC protocols. These emission data may not represent real-use exposure to harmful or potentially harmful constituents. Recent advances in topography capture methods enable smoking behaviors to be quantified in the natural environment, which can be used to inform emissions study protocols.

The intent of the current work was to capture a range of real smoking behaviors and to explore how well the FTC/ISO (non-intense), MDPH (medium), and HC (intense) protocols represent the observed behaviors. The hypothesis was that natural environment observations of cigarette smokers’ topography would exhibit a wide range of mean puff volumes, durations, and flow rates beyond what is represented by the commonly used puffing protocols. This work is the first step in developing machine-puffing protocol standards that better reflect actual smoking behavior and that will lead to more meaningful risk assessments.

## 2. Materials and Methods

### 2.1. Participants and Recruitment

Participants were current regular smokers over the age of 18 years. Participants were recruited from the Rochester Institute of Technology (RIT) campus and the greater Rochester community. Participants were recruited using mass emails sent to the campus distribution list, in conjunction with flyers posted around the campus and with social media Facebook ads between April and November 2018. Advertising referred to a research study regarding cigarette smoking and stated that participants may be eligible to receive $50 for participating in a 1-week observation period if they were over the age of 18 and were current cigarette users.

### 2.2. Instruments and Variables

Each participant was provided with a wPUM™ cigarette topography monitor, as previously described [17], with a disposable mouth pipe designed to fit varying diameters of cigarettes, to use while smoking their normal brand of cigarettes in their natural environment for one week. The monitor is a hand-held device with an ergonomic finger-grip shape into which study participants insert the filter-end of their cigarette prior to smoking. The wPUM^TM^ records the flowrate, duration, and volume of every puff taken, along with a date and time for every puff.

Prior to each participant’s intake appointment, a technician conducted a pre-deployment flow-rate calibration of the wPUM™ cigarette monitor anticipated to be assigned to that participant [18]. Flow-rate calibration was done using the fully characterized RIT PES-1™ calibration system [19], which employs flow meters certified annually by a third-party vendor. Each monitor was cleaned, calibrated, and readied with a new primary-cell battery and a formatted data-storage card. Each monitor was calibrated post-deployment for each participant, to verify the consistency of monitor performance.

### 2.3. Procedures

The study protocol was reviewed and approved by the Rochester Institute of Technology (RIT) Institutional Review Board (IRB). The human subject’s natural environment puff topography study protocol consisted of (1) online pre-screening, (2) an intake appointment, (3) a monitoring period, and (4) an outtake appointment. Each step is described below.

Pre-screening was done using an online pre-screening survey intended to identify and exclude individuals who did not meet the eligibility requirements. The PhenX Tobacco Use Survey was used, which is an instrument available in the PhenX Toolkit [20] designed to assess the history and frequency of use of a wide range of tobacco products. Individuals passed the pre-screening if their responses indicated that they consented to participate, were at least 18 years of age, and were current users as defined by the PhenX survey. Specifically, participants were asked, “Do you now smoke cigarettes every day”, with potential answers being “some days”, “not at all”, and “don’t know or refuse”. If they answered “every day” or “some days”, they were considered current users. The research administrator responded to each eligible survey respondent with detailed information about the study and invited them to make an intake appointment. Individuals who did not pass the pre-screening were notified immediately after taking the online survey.

Intake appointments took place in the Respiratory Technologies Lab (RTL) and were made on a first-come, first-serve basis. Each appointment lasted between fifteen minutes to one hour and included a final screening to confirm that the participants met the inclusion/exclusion criteria. The inclusion criteria included: having no current respiratory illness or disease, not being pregnant nor intending to become pregnant during the observation period, identifying as current smokers, having their own cigarettes to use during the study, and being able to verify their age via government-issued identification. All persons who attended the intake appointment were offered information on smoking cessation, including resources for quitting, and were asked if they were interested in quitting. Participants who were not interested in quitting met the inclusion criteria and signed the informed consent were enrolled in the study. Participants who signed the informed consent were provided with a pre-calibrated wPUM™ cigarette monitor, a daily study log to record cigarette usage, and a study packet describing the study protocol and monitor operation. Participants were instructed on the proper use of the monitor and given an opportunity to turn it on and off in the lab. Participants were invited to contact the research administrator during the observation period if they encountered any difficulties. Participants were asked to schedule an in-lab outtake appointment with the research administrator via email, and then dismissed.

The monitoring period began immediately after the intake appointment concluded and lasted for one week. The study protocol was designed to begin and end on a Tuesday, Wednesday, or Thursday to capture weekday and weekend behavior without interruption. Participants were instructed to smoke as they normally would throughout their day-to-day activities, using their usual brand of cigarettes in conjunction with the cigarette monitor for every cigarette smoked until they returned for their outtake appointment. Participants were asked to record the cigarette brands they used each day on their daily study log. They were also asked to record instances when they smoked cigarettes without the monitor.

At the conclusion of the monitoring period, participants came to the RTL for their outtake appointment. During the outtake appointment, participants returned the monitor and their daily study log of cigarette usage to the research administrator and participated in an exit interview to assess product and monitor use during the observation period and to identify difficulties encountered during the study. The research administrator confirmed that the participants completed the online questionnaires emailed to them the day prior to their outtake appointment. These included a nicotine dependence questionnaire (NDQ) [20,21] and an online exit questionnaire that asked participants about their experience using the monitor. Once the exit interview was completed, participants were compensated for their participation and dismissed.

### 2.4. Data Analysis

Monitors were tested for repeatability using a pre- and post-calibration protocol, before and after deployment for each participant. Each file captured with the cigarette monitor was evaluated by an analyst to identify the presence of flow-path contaminants, which may impede subsequent data analysis. The participant study logs were checked to assess compliance to the protocol. Automated topography analysis of raw monitor data was conducted by the TAP^TM^ topography analysis program for every puffing session of every participant, converting noisy raw voltage into clearly identified discrete puffs with known duration, mean flow rate, puff volume, and interpuff interval (Figure 1) as previously described [22]. The TAP^TM^ program puff detection criteria was set to a minimum puff flow rate of 10 mL/s and a minimum interpuff interval of 0.2 s. The TAP program calculated the topography parameters including puff count, mean puff flow rate, duration, volume, and interpuff interval for every puff taken by all participants over the course of the week-long observation.

Descriptive statistics were presented for the puff flow rate, duration, volume, and interpuff interval of each participant and the cohort. Confidence intervals on the mean and median, interquartile range, and outlier analysis were conducted on the data and presented for comparison with the deterministic FTC, MDPH, and HC puff profiles. We tested the null hypothesis that there is no difference between the deterministic puff volume and duration of each commonly used puffing profile with each observed individual and with the group mean of study participants. The effect of nicotine dependence variation among participants was assessed using multivariate linear regression analysis on all topography parameters.

## 3. Results

### 3.1. Cohort Demographics

A total of 117 replies were received from prospective participants in response to the recruitment announcements. Figure 2 illustrates the management of all 117 replies. From the online pre-screening, 46 participants were invited to an intake appointment, and 28 individuals were enrolled in the study. From these, 27 participants completed the study. Data from one participant was excluded during data integrity checking due to a broken monitor. The participant noted in the study log that the monitor had been dropped. Data from *n* = 26 study participants are presented herein. One participant reported on the daily study log having smoked marijuana through the wPUM monitor on 2 of the observation days, in addition to smoking. Since we could not distinguish which sessions were marijuana and which were cigarettes, the topography from those two days was not included in that participant’s statistics; however, both sets of data are included in the Appendix A. The supplemental data file includes demographics for each participant, along with cigarette brands used; results from the tobacco use survey related to cigarette, electronic cigarette (ecig), and hookah use; and NDQ scores for cigarettes and ecigs.

The sample of *n* = 26 participants was composed of 23 males and 3 females ranging in age from 20 to 56 years with a mean (STD) age of 27 (7) years. Of this cohort, 54% self-identified as Asian and 35% as white. Two participants self-identified as black or African American, one as Hispanic or Latino; and one participant preferred not to report their race. Participants reported using the following cigarettes in their study logs: L&M (Blue) 100s, Newport 100s, Marlboro Gold, Marlboro Gold 100s, Marlboro Red, Marlboro Red 100s, Marlboro Silver, Seneca Full Flavor, Seneca Silver, Seneca Light 100s, American Spirit Yellow, Camel Menthol Crush, and Camel Silver. Some participants reported using more than one brand.

Based on responses to the PhenX tobacco use survey, 25 participants were regular current smokers (>50 lifetime cigarettes) and 1 was an experimental current smoker (≤50 lifetime). Of the regular current smokers, 24 were everyday smokers and 1 was a “some days” smoker. Current dual use of ecigs was reported by eight participants (all but 1 were experimental current ecig users); and current dual use of hookah was reported by six participants (all but two were experimental current hookah users). Former ecig use was reported by 10, and former hookah use was reported by 11.

Based on the cigarette NDQ [20,21], nicotine dependence scores for cigarettes ranged from 1 to 16, with a mean (STD) of 7.9 (4.2). The cigarette NDQ concluded that 3 participants had no dependence (one of these participants was the “some days” user), 12 had low dependence, 8 had medium dependence (one of these participants was the experimental current user), and 3 had high dependence. Based on the ecig NDQ [23], 2 participants had low dependence on ecigs, and the other 24 participants had no dependence on ecigs.

### 3.2. Cigarette Smoking Natural Environment Topography

A total of *n* = 8250 puffs were measured across 26 smokers monitored for 1 week in their natural environment. Mean topography parameters for the cohort are given in Table 1, including puff count, flow rate, duration, volume, and the interval between puffs. Descriptive statistics for each participant are given in the supplemental data file, and illustrated by box plots in Figure 3. A relatively wide intra- and intersubject variability was observed, suggesting that a single puffing protocol is insufficient to represent real-use behavior. A scatter plot of the mean volume versus duration for each participant is shown in Figure 4 and illustrates the extent to which the non-intense, medium-, and intense-smoking protocols represent the observed smoking behavior. The FTC, MDPH, and HC protocols are represented by the yellow, green, and purple markers, respectively. The blue markers show the means with 95% confidence intervals for duration and flow rate for each participant in their natural environment. The black marker presents the mean topography with 95% confidence intervals for the study cohort. None of the three standard protocols overlap with the 95% confidence interval of any single participant, nor with the 95% confidence interval of the group cohort mean. We conclude with high confidence, *p* < 0.001, that none of the FTC, MDPH, or HC protocols realistically reflect the mean behavior of individual participants or the cohort. Therefore, the protocols are insufficient to reflect puffing topography representative of emissions that would be experienced by smokers in their natural environments. Multivariate linear regression analysis indicated nicotine dependence did not correlate with puff volume (*p* = 0.84), puff duration (*p* = 0.94), or puff flow rate (*p* = 0.98). Nicotine dependence correlated somewhat with puff count (*p* = 0.095).

## 4. Discussion

The working hypothesis for this study was that natural environment observations of cigarette smokers’ topography would exhibit a range of topography parameters beyond what is represented by the commonly used puffing protocols. We rejected the null hypothesis that there was no difference between the observed individual or group mean of study participants and the deterministic puff volume and duration of any of the three commonly used puffing protocols. None of the protocols, FTC/ISO, MDPH, nor HC, individually or collectively represented the mean or range of smoking behavior exhibited in the natural environment.

The results are sufficient for assessing whether the commonly used puffing protocols reflect the topography exhibited by the study cohort. The sample of participants spans a range of brands, dual tobacco-product-use categories, demographics, and nicotine dependence. This random sample was chosen to reflect the fact that the commonly used puffing protocols are not brand specific and make no mention of the population demographics or daily use frequency to which the protocols apply.

The results have implications regarding the FDA’s current draft guidance for capturing HPHC emissions. The HC and FTC/ISO protocols, which were recommended by the FDA for intense and non-intense puffing, respectively, fall short of capturing the extreme behaviors present in our cohort, suggesting new FDA recommendations are needed. Further, results indicate that smoking intensity is defined too narrowly and must involve more than just puff duration. Smoking intensity can also involve larger puff volumes, shorter interpuff intervals, or faster flow rates. Puff flow rate, duration, and interpuff interval can each mechanistically affect the way that cigarettes burn and thus can affect the concentration of HPHCs generated. A smoker could take a small-volume puff that was highly concentrated with HPHCs, due to the manner in which their puffing affected the burn rate. Thus, referring to a low puff volume as non-intense puffing may lead to misguided conclusions. Topography parameters are interrelated and must be considered holistically when devising a protocol that encompasses the intensity range of possible behaviors.

The topography data presented here should not be considered generalizable to the population of smokers. The results presented herein do not address the effect of race/ethnicity, age, nicotine dependence, consumption patterns, dual-use of multiple tobacco products, behavioral factors, mental health, membership in underrepresented groups, socioeconomic status, or other confounders of the topography characteristics of these subpopulations. Given the impact of topography on emissions and exposure to HPHCs, this work points to the need for additional studies to assess topography of key subgroups, and to evaluate the validity of current and proposed smoking emissions puff protocols relative to each group.

## 5. Conclusions

This paper presented cigarette topography measured in the natural environment over the course of one week for a range of smokers, cigarette brand choices, nicotine dependency, and dual product usage. Data suggest current commonly used puffing protocol standards are insufficient to test cigarettes across the wide range of real-use behaviors. Results have implications on emissions studies and on FDA guidance, which currently relies on a limited set of topography standards. More work is needed to develop better standards for machine-puffing protocols such that resulting benchtop emissions better reflect actual smoking behavior and lead to more meaningful risk assessments.

## Figures and Tables

**Figure 1 ijerph-17-03444-f001:**
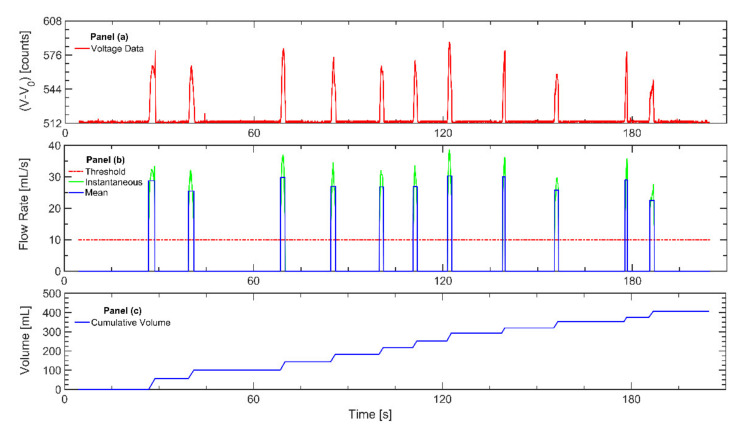
The figure illustrates the process for analyzing data recorded with the wPUM™ cigarette monitor after each monitor is returned to the lab: (**a**) The raw voltage data (red line) is extracted from the monitor. (**b**) The voltage data are converted to instantaneous flow rate (green line) using the monitor-specific calibration curve. A minimum threshold of 10 mL/s (red line) is used to identify the onset and conclusion of puffs. The mean flow rate of each puff is determined (blue line). (**c**) The mean puff flow rate is integrated to estimate the cumulative session volume (blue line).

**Figure 2 ijerph-17-03444-f002:**
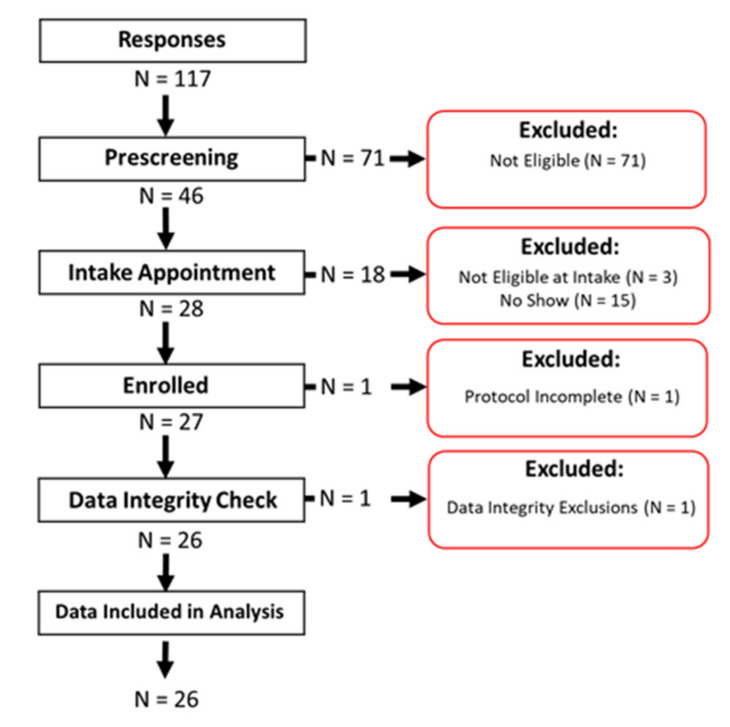
Cohort study flow chart, where *n* = 117 initial responses were received, *n* = 46 respondents were found eligible and invited to an intake appointment, and *n* = 28 participants were enrolled. One participant withdrew before completing the one-week protocol, and one participant’s data set was excluded during the data integrity check due to a broken monitor. Data from *n* = 26 participants were included in the data analysis and are presented here.

**Figure 3 ijerph-17-03444-f003:**
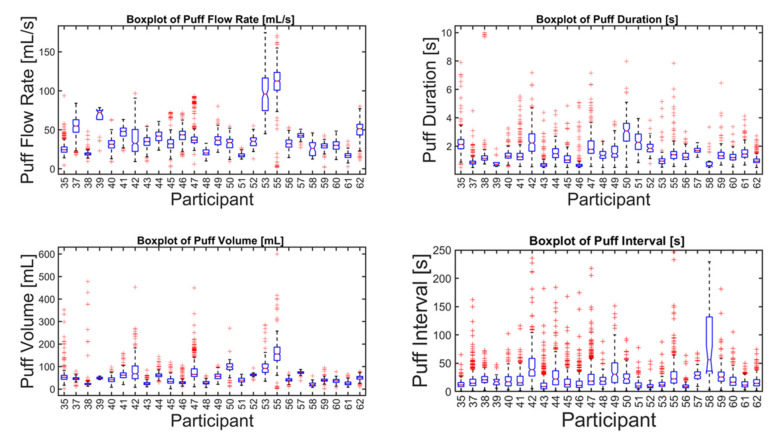
Descriptive statistics of puff flow rate, duration, volume, and interval of (*n* = 26 smokers) cigarette smokers in their natural environment during a week-long observation period (*n* = 8250 puffs). The box plot for each parameter and participant indicates the median (50th percentile, red horizontal line), 95% confidence interval on the median (box notches), 25th and 75th percentiles (lower and upper edge of boxes, respectively), lower and upper fences or whiskers (vertical lines representing 1.5 times the interquartile range), and data outliers (red markers).

**Figure 4 ijerph-17-03444-f004:**
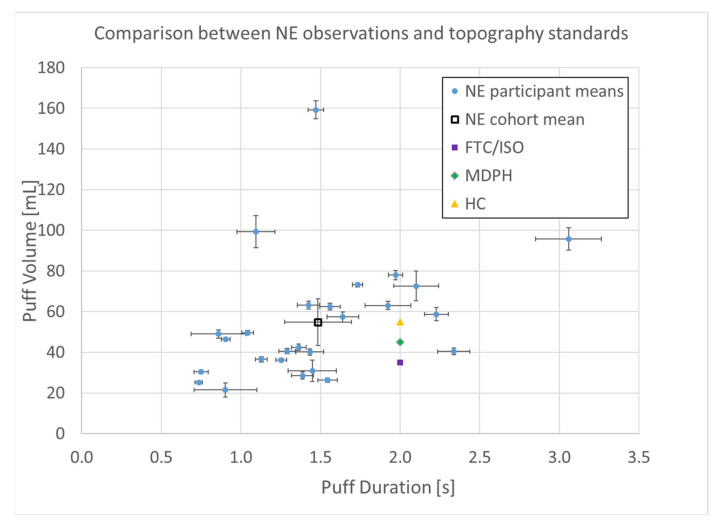
Puff topography distribution (*n* = 8250 puffs) of (*n* = 26 smokers) cigarette smokers in their natural environment (NE) during a week-long observation period. Shown are means with 95% confidence intervals.

**Table 1 ijerph-17-03444-t001:** Cohort mean descriptive statistics compared to currently available standards.

	Puff Count for the week	Puff Flow Rate (mL/s)	Puff Duration (s)	Puff Volume (mL)	Puff Interval (s)
Natural Environment					
*n*	26	26	26	26	26
Mean	314	40.4	1.5	54.9	32.5
STD	233	21.7	0.5	29.8	29.4
SEM	46	4.3	0.1	5.8	5.8
Min	18.0	17.2	0.7	21.4	10.2
Max	994	110.6	3.1	159.2	140.2
Standards					
FTC/ISO	---	17.5	2	35	60
MDPH	---	22.5	2	45	30
HC	---	27.5	2	55	30

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
