# Peer review of "A Comparison between Cigarette Topography from a One-Week Natural Environment Study to FTC/ISO, Health Canada, and Massachusetts Department of Public Health Puff Profile Standards"

_ijerph, 2020, doi:10.3390/ijerph17103444_

Round 1

Reviewer 1 Report

Thank you for giving me the opportunity to review the manuscript titled “Combustible Cigarette Topography from a One Week Natura Environment Study.” The topic addressed is quite interesting. However, I have major concerns about the rest of the manuscript that limits my enthusiasm.

Introduction

 The readability of this section should be further improved. The introduction is made of short sentences that do not connect similar ideas.

The authors fail to summarize what it is known about each protocol (advantages, disadvantages, et)

The sentence “further suggests that each brand has its own topography” is ambiguous. Smokers have different topography patters. Cigarettes have different characteristics that may influence smoker’s topography.

Certain sections of the introduction are difficult to understand. The authors use terms and references that are not common to the whole scientific community (e.g. 77FR200300, ISO, etc).

The rationale of conducting this study is not discussed nor introduced adequately (why is important this research, what novel information provides, etc.).

Methods

I strongly recommend the authors restructure this section as participants, instruments and variables, procedure, and data analysis.

The authors indicated in the Introduction that each cigarette has its own topography. However, the authors did not standardize the type of cigarette used across participants. Is any reason for this?

Specific instruments used from the Phenx Tobacco survey should be described.

Statistical analyses conducted are missing

Results

Conclusion

The discussion needs major changes. It is unclear which are the main findings of this paper.  I recommend the authors identify the main and secondary outcomes; then, present each of these findings in the first paragraph of the discussion; and, finally explain each finding separately.

There is a large portion of information that is repeated from the Results section.

The hypothesis should be presented in the Introduction as well.

The implications of this research should be further discussed.

Other comments

Participants enrolled in this study have very different characteristics (age, dependence, type of smoker, use of other tobacco products). This wide variability seems to limit the applicability of these findings to any specific profile of smokers.

Reviewer 2 Report

Overall, the manuscript is an interesting study in an exciting area of tobacco research. Strengths of this study include its naturalistic design and application. Several limitations are acknowledged by the authors, including the confound of other product use and dual use. However, additional issues must be addressed to improve the overall impact of this study and to increase the overall readability. Specific items are listed below. Thank you for the opportunity to review this interesting research.

  • The title should more accurately reflect the objectives of the study. Based on the introduction, the study has a narrow focus on evaluating the fit of the three testing protocols to inform standards for machine puffing protocols. This focus was unexpected based on the current title.
  • The first sentence of the abstract discusses cigarette testing protocols and intensity. However, no background has been given on these protocols. I suggest starting more broadly and introducing more specialized concepts in the introduction.
  • There are a number of acronyms in the abstract that are not explained.
  • Please ensure all acronyms are defined. Similarly, I suggest removing any unnecessary acronyms, particularly in the introduction, to increase readability.
  • The “recruitment” section should be relabeled to “participants and recruitment” or something similar to reflect that the study population is also discussed.
  • Were ineligible participants (based on pre-screener) able to complete the pre-screener again? If so, it is possible that a motivated individual could figure out how to be eligible based on the screener. How did the authors protect against this?
  • Related to this, the authors state that final eligibility was confirmed at the intake appointment. How was this confirmed? Specifically, was CO measured or were participants asked to bring and show their cigarettes or were all items confirmed verbally?
  • Page 3, line 111: “user” should be changed to “use”
  • Please clearly list all inclusion/exclusion criteria. Most of it can be inferred from sections 2.2.2 and 2.2.3 but a clear list would improve clarity.
  • In section 2.2.5, the authors state that participants completed a log of e-cigarette dependence. However, this is the first mention of e-cigarette use. Were other tobacco/marijuana/e-cigarette users allowed into the study? If so, this should be reported clearly in the methods and a section added to the discussion regarding how this may have impacted their cigarette smoking topography.
  • I am not familiar with the term “outtake appointment”. One sentence stating when this appointment occurred and the purpose of it would improve clarity.
  • Many participants were using products other than cigarettes. This appears to be a significant confound and should be discussed further.
  • The authors recruited both daily and non-daily smokers into the study.
  • I am surprised that 12 participants had no dependence despite the majority of participants reporting daily smoking. Is this a problem with the measure used to assess dependence or another issue?
  • According to the puff count data, participants were smoking 4.4 cigarettes per day, indicating many of these participants are very light smokers. In addition, at least some of the participants were non-daily smokers. This raises concerns regarding the generalizability of these data. This issue should be discussed.
  • Similarly, it begs into question whether these data, given the small N and heterogeneity of participants, are appropriate for evaluating the outlined protocols/standards. The study is interesting and important but may not be appropriate for evaluation of these standards.
  • What does “NE” mean in figure 4? Natural environment?
  • Page 7, line 226: “smoker’s” should be “smokers’”
  • The racial makeup of the sample should be discussed in the discussion. Specifically, data suggests that African American smokers smoke fewer cigarettes per day but have greater morbidity and mortality due to differential puff topography. While it is interesting that so many participants identified as Asian, the current data are likely not generalizable to other groups (including AA smokers). This warrants discussion.

Round 2

Reviewer 2 Report

The authors addressed my comments and I have no further revisions to suggest. Thank you for the opportunity to review this interesting study.